# Hospital Admission Due to Hypotension in Australia and in England and Wales

**DOI:** 10.3390/healthcare11091210

**Published:** 2023-04-23

**Authors:** Sara Ibrahim Hemmo, Abdallah Y. Naser, Esra’ O. Taybeh

**Affiliations:** Department of Applied Pharmaceutical Sciences and Clinical Pharmacy, Faculty of Pharmacy, Isra University, Amman 11622, Jordan

**Keywords:** admission, Australia, England, hypotension, Wales

## Abstract

Objectives: Hypotension is overlooked because it is often harmless, easily reversed, and can have few or even no symptoms. However, complications of untreated hypotension are dangerous and can result in death. The aim of this study was to examine the trend of hospital admission due to hypotension in Australia and in England and Wales between 1999 and 2020. Method: This was a secular trend analysis study that examined the hospitalisation pattern for hypotension in Australia, England, and Wales between 1999 and 2020. Hospitalisation data were obtained from the National Hospital Morbidity Database in Australia, Hospital Episode Statistics database in England, and Patient Episode Database for Wales. We analysed the variation in hospitalisation rates using the Pearson chi-square test for independence. Results: Our study showed that hypotension hospital admission rates increased significantly between 1999 and 2020 by 168%, 398%, and 149% in Australia, England, and Wales, respectively. The most common hypotension hospital admissions reason was orthostatic hypotension. All types of hypotension-related hospital admissions in Australia, England, and Wales were directly related to age, more common among the age group 75 years and above. Bed-days hypotension hospital admission patients accounted for 84.6%, 99.5%, and 99.7% of the total number of hypotension hospital admissions in Australia, England, and Wales. Conclusion: In the past two decades, orthostatic hypotension was the most prevalent type of hypotension that required hospitalization in Australia, England, and Wales. Age was identified as the primary risk factor for hypotension across all causes. Future research should focus on identifying modifiable risk factors for hypotension and developing strategies to reduce the burden of orthostatic hypotension.

## 1. Introduction

With regards to blood pressure, hyper- rather than hypotension gets most of the attention in the scholarly literature [1]. This is not surprising since hypertension affects more than 30% of the adult worldwide population [2] and contributes to almost 1000 deaths per day [3]. Hypotension, on the other hand, is overlooked because it is often harmless, easily reversed, and can have few or even no symptoms. However, complications of untreated hypotension are dangerous and can result in death [4].

Hypotension is defined as a decrease in systemic arterial blood pressure and can be symptomatic or asymptomatic [4,5,6,7]. Though there is no accepted standard value, hypotension is considered when the blood pressure is less than 90 millimetres of mercury (mm Hg) for systolic and\or less than 60 mm Hg for diastolic [5,6,7].

Hypotension signs and symptoms include blurred or fading vision, fatigue, light-headedness or dizziness, lack of concentration, nausea, and fainting [5,6,7]. Moreover, severe hypotension can cause shock, which is a life-threatening condition that includes the following signs and symptoms: shallow and rapid breathing; confusion [particularly in senior people]; pale, clammy, and cold skin; and fast and weak pulse [7]. One of the main causes for hypovolemic shock is bleeding, which leads to a significant fluid loss [8]. Untreated hypotension can lead to multiple organ failure in the case of fulminant shock or impending shock or to death in hypotensive patients with severely impaired cardiac output [4].

Early detection and immediate therapeutic intervention are the keys to efficient hypotension management. The most crucial aspect of treating hypotension is treating its causes [9]. Orthostatic hypotension is the most common form of low blood pressure [1]. Some researchers have found relations between hypotension and various antihypertensive drugs [10]. A significant reduction in blood pressure can result from excessive or off-label use. Orthostatic hypotension is also associated with different comorbidities such as diabetes mellitus and Parkinson’s disease, or it may result from iatrogenic complications while the patient is admitted to the hospital or maybe is the reason for the hospital admission [11]. Hospital admissions for hypotension increase mortality from cardiovascular diseases (CVD)s [12]. A previous prospective observational study found that, compared with individuals with normal blood pressure aged 75 years and over, mortality rates were increased in those with hypotension [13]. Hospital mortality rates also increase due to non-traumatic hypotension outside the hospital [11]. Furthermore, hypotension is common in the emergency department and linked with an elevated mortality rate [14].

Since patients with hypotension are prone to severe complications, including irreversible end-organ damage, and eventually death [4], physicians must determine the clinical manifestation of patients with reversible or life-threatening reasons for hypotension and apply appropriate therapeutic intervention. Therefore, it is necessary to assess the extent of this health issue by identifying the current trend of hypotension-related hospital admissions to help them develop strategies of care resulting in reducing and preventing this health issue. Accordingly, this research aims to study the trend of hospital admission due to hypotension in Australia and in England and Wales between 1999 and 2020.

## 2. Methods

### 2.1. Data Sources

#### 2.1.1. Hospital Episode Statistics Database and the Patient Episode Database for Wales

Hospital Episode Statistics database (HES) database in England and Patient Episode Database for Wales (PEDW) databases record all hospital admissions, Outpatients, and Accident and Emergency (A&E) activities performed at all National Health Service (NHS) trusts and any independent sector funded by NHS trusts. Data for hospital admissions in England and Wales are available from the years 1999/2000 onwards. Available data include patient demographics, clinical diagnoses, procedures, and duration of stay. HES and PEDW data are checked regularly to ensure their validity and accuracy [15,16].

#### 2.1.2. National Hospital Morbidity Database

The National Hospital Morbidity Database (NHMD) is part of the National Hospitals Data Collection. National Hospitals Data Collection includes the major national hospital databases held by the Australian Institute of Health and Welfare (AIHW) [17]. NHMD is an online database in which data provided by state and territory health authorities in Australia are collected [18]. The data collected at the NHMD are sets of episodes-level records from the morbidity data collection systems of patients admitted to private and public hospitals in Australia. 

#### 2.1.3. Office for National Statistics (ONS)

Office for National Statistics (ONS) was used to collect England and Wales mid-year population data for the period between 1999 and 2020 [19].

#### 2.1.4. Australian Bureau of Statistics

The Australian Bureau of Statistics (ABS) is the Australian National Statistics Agency and the official source for reliable and independent information [20]. ABS and the national, state, and territory population data was used to collect mid-year population data for the period between 1998 and 2020 [21,22].

### 2.2. Study Population

All publicly available hypotension admissions data from the HES database [15] and PEDW for the period between April 1999 and April 2020 were included in the study [23]. All private and public hypotension admissions data from 1999 to 2020 in Australia were included in the study. That comprises all available data from NHMD in principal diagnosis data cubes [24]. The following ICD codes were used to extract the data for this study: I95.0, I95.1, I95.2, I95.8, and I95.9.

### 2.3. Statistical Analysis

SPSS version 27 (IBM Corp., Armonk, NY, USA) was used for all analyses. Hospitalisation rates with 95% CIs were determined by dividing hospitalisation episodes by the mid-year population. A bed-day (or inpatient day) is defined as “a day during which a person admitted as an inpatient is confined to a bed and in which the patient stays overnight in a hospital”. Overnight-stay admitted care is defined as “the care provided for a minimum of one night, to a patient who is admitted to and separated from the hospital on different dates”. Pearson chi-square test for independence was used to analyse the variation in hospitalisation rates between 1999 and 2020.

## 3. Results

### 3.1. Trends in Total Hypotension Hospital Admissions

In Australia, the total annual number of hypotension hospital admissions for different causes increased by 261.7% from 4848 in 1999 to 17,533 in 2020, representing an increase in hospital admission rate of 167.8% [from 25.48 (95% CI 24.76–26.19) in 1999 to 68.24 (95% CI 67.23–69.25) in 2020 per 100,000 persons, *p* < 0.01]. In England, the total annual number of hypotension hospital admissions for different causes increased by 471.7% from 8677 in 1999 to 49,608 in 2020, representing an increase in hospital admission rate of 397.8% [from 17.62 (95% CI 17.25–18.00) in 1999 to 87.72 (95% CI 86.95–88.50) in 2020 per 100,000 persons, *p* < 0.001]. In Wales, the total annual number of hypotension hospital admissions for different causes increased by 171.7% from 671 in 1999 to 1823 in 2020, representing an increase in hospital admission rate of 149.2% [from 23.08 (95% CI 21.34–24.83) in 1999 to 57.52 (95% CI 54.88–60.15) in 2020 per 100,000 persons, *p* < 0.01] (Figure 1).

The most common hypotension hospital admissions reason in England, Wales, and Australia was orthostatic hypotension which accounted for 74.7%, 73.5%, and 61.0%, respectively (Table 1).

### 3.2. Trends in Total Hypotension Hospital Admissions Stratified by Hospital Stay

In Australia, over-night hypotension hospital admission patients accounted for 84.6% of the total number of hypotension hospital admissions, and 15.4% were same-day patients. Rates of same-day hospital admission for hypotension increased by 294.4% [from 2.86 (95% CI 2.62–3.10) in 1999 to 11.28 (95% CI 10.87–11.69) in 2020 per 100,000 persons]. Rates of overnight-stay hospital admission for hypotension increased by 151.9% [from 22.61 (95% CI 21.94–23.29) in 1999 to 56.96 (95% CI 56.03–57.88) in 2020 per 100,000 persons]. In England, bed-days hypotension hospital admission patients accounted for 99.5% of the total number of hypotension hospital admissions, and 0.5% were day-case patients. Rates of day-case hospital admission for hypotension increased by 126.9% [from 0.78 (95% CI 0.70–0.86) in 1999 to 1.77 (95% CI 1.66–1.88) in 2020 per 100,000 persons]. Rates of bed-days hospital admission for hypotension increased by 98.6% [from 135.39 (95% CI 134.36–136.42) in 1999 to 268.83 (95% CI 267.48–270.18) in 2020 per 100,000 persons]. In Wales, bed-days hypotension hospital admission patients accounted for 99.7% of the total number of hypotension hospital admissions, and 0.3% were day-case patients. Rates of day-case hospital admission for hypotension increased by 660.0% [from 0.10 (95% CI −0.01–0.22) in 1999 to 0.76 (95% CI 0.45–1.06) in 2020 per 100,000 persons]. Rates of bed-days hospital admission for hypotension increased by 69.3% [from 166.47 (95% CI 161.78–171.15) in 1999 to 281.77 (95% CI 275.94–287.61) in 2020 per 100,000 persons] (Figure 2).

### 3.3. Trends in Total Hypotension Hospital Admissions Stratified by Age Group

Hypotension-related hospital admissions were directly related to age (more prevalent among the age group 75 years and above), which accounted for 63.2%, 70.4%, and 67.4% of the total number of hypotension hospital admissions in Australia, England, and Wales, respectively (Table 2).

Hospital admission rates for hypotension among patients aged below 15 years increased by 13.5%, 234.8%, and 109.3% in Australia, England, and Wales, respectively. Hospital admission rates for hypotension among patients aged 15–59 years increased by 185.5%, 323.2%, and 122.2% in Australia, England, and Wales, respectively. Hospital admission rates for hypotension among patients aged 60–74 years increased by 96.8%, 273.8%, and 64.6% in Australia, England, and Wales, respectively. Hospital admission rates for hypotension among patients aged 75 years and above increased by 111.0%, 360.9%, and 134.4% in Australia, England, and Wales, respectively (Table 3, Figure 3).

### 3.4. Trends in Total Hypotension Hospital Admissions Stratified by Gender

From 1999/2000 to 2019/2020, there were 212,890 admissions in Australia (50.8% for males), 554,793 admissions in England (50.3% for males), and 29,458 in Wales (50.0% for males). Hypotension hospital admission rate among males increased by 207.8%, 498.6%, and 215.1% in Australia, England, and Wales, respectively (Table 4, Figure 4).

Hypotension hospital admission rate among females increased by 134.0%, 316.7%, and 97.7% in Australia, England, and Wales, respectively (Table 4, Figure 5).

### 3.5. Trends in Types of Hypotension Hospital Admissions

#### 3.5.1. Idiopathic Hypotension

From 1999/2000 to 2019/2020, a massive increase in idiopathic hypotension hospital admissions rate was observed in England and Wales with 178.3% and 419.7%, respectively. An increase in idiopathic hypotension hospital admissions rate was also seen in Australia by 80.5% (Table 5, Figure 6).

#### 3.5.2. Orthostatic Hypotension

From 1999/2000 to 2019/2020, an increase in orthostatic hypotension hospital admissions rate was observed in the three countries, Australia, England, and Wales, with 228.6%, 460.6%, and 195.6%, respectively (Table 5, Figure 6).

#### 3.5.3. Hypotension Due to Drugs

From 1999/2000 to 2019/2020, a noted increase in hypotension due to drugs hospital admissions rate was observed in England by 212.7%. Additionally, an increase in hypotension due to drugs hospital admissions rate was observed in Australia by 28.6%, while a decrease trend was observed in Wales by 17.8% (Table 5, Figure 6).

#### 3.5.4. Other Hypotension

An increase in other hypotension hospital admissions rates was observed in England and Wales by 470.4% and 122.7%, respectively, whilst a decrease in other hypotension hospital admissions rates was detected in Australia by 16.7% (Table 5, Figure 6).

#### 3.5.5. Unspecified Hypotension

Unspecified hypotension hospital admissions rate was increased in England and Australia with 262.3% and 118.3%, respectively. In the same period, an increase in hospital admissions rate due to unspecified hypotension in Wales with 85.8% (Table 5, Figure 6).

#### 3.5.6. Trends in Types of Hypotension Hospital Admissions Stratified by Age Group

All types of hypotension-related hospital admissions in Australia, England, and Wales were directly related to age, more common among the age group 75 years and above (Figure 7).

### 3.6. Trends in Types of Hypotension Hospital Admissions Stratified by Gender

Hypotension hospital admission rates in England were higher among males compared to females, except for idiopathic hypotension, which was more prevalent among females (Figure 8). Hypotension hospital admission rates in Australia were also higher among males compared to females, except for idiopathic hypotension and orthostatic hypotension which were more prevalent among females (Figure 8). In Wales, hypotension hospital admission rates for idiopathic hypotension and other hypotension were more prevalent among females (Figure 8).

## 4. Discussion

The present study brings to light the hospitalization pattern for hypotension in three developed countries; Australia, England, and Wales during the period between 1999 and 2020. Our study showed that hypotension hospital admission rates increased significantly during the study period by 168%, 398%, and 149% in Australia, England, and Wales, respectively. Several factors may contribute to the variance in the increased rate of hospital admissions due to hypotension in Australia, England, and Wales, including population growth, age structure, gender-based ratio, life expectancy, and the burden of hypertension disease [19,20,21,22]. For, instance, the total population increased by 35.0%, 14.9%, and 9.0% in Australia, England, and Wales, respectively, from mid-2000 to mid-2020 [19,20,21,22]. Moreover, the percentage of the population aged below 15 years and those aged 15–59 years from the total population decreased in the three studied countries, while the percentage of the older population increased by 34.4%, 17.5%, and 22.7% for those of ages 60–74 and by 27.7%, 15.2%, and 18.1% for those of ages 75 years and older in Australia, England, and Wales, respectively. On the other hand, a slight change in the percentage of the male population in Australia, England, and Wales was observed [from mid-2000 increased by 1% in England, 2% in Wales, and decreased by 0.1% in Australia] [19,21,22]. Furthermore, life expectancy has increased by 4.9% in Australia [from 79.3 years in 1999 to 83.4 years in 2019], and by 4.7% in the UK [from 77.5 years in 1999 to 81.3 years in 2019] [25]. Additionally, the cardiovascular diseases accounted in 2019 for 12.6%, 14.6%, and 16.4% of total diseases burden in Australia, England, and Wales, respectively [26]. Previous statistics showed that the increase in the admissions related to hypotension is not consistent with the change in gender ratio or the increase in the life expectancy, but this could be due to the increase in the prevalence of the condition itself [19,21,22,25,26].

The reasons for the variation in hypotension admission rates include the use of certain medications [such as those for hypertension, erectile dysfunction, Parkinson’s disease, or depression], dehydration, blood infection or blood loss, pregnancy, cardiac diseases [such as heart failure, bradycardia, heart attack, tachycardia, or heart valve problems], extreme changes in body temperature, endocrine problems including hypoglycaemia, Addison’s disease, or parathyroid disease, and intense emotions [such as fear or pain] [5,7,27].

Our study found that the most common hypotension hospital admissions reason in Australia, England, and Wales was orthostatic hypotension. That is in line with a previous study that found orthostatic hypotension to be the most common form of hypotension [28]. In addition, estimates of the prevalence of orthostatic hypotension range widely, from 5% to 30% for individuals aged 50 years and below to those over 70 years, respectively [29]. Orthostatic hypotension enormously increases risks for mortality, cardiovascular disease, dementia, and falls [30]. Regardless of orthostatic hypotension aetiology, it is the second most prevalent reason for syncope. It also results in more than 50% hospitalization among patients aged over 80 years, possibly the most severe hemodynamic outcome [31].

Our findings showed that all types of hypotension-related hospital admissions in Australia, England, and Wales were directly related to age, more common among the age group 75 years and above. In Australia, hospital admissions rates for cardiovascular disease (CVDs) increased with age, with over 83% CVDs hospital admissions arising among patients aged 55 and above in 2019 [32]. This could be due to the fact that the prevalence of most CVDs, including hypotension, is much higher among the older population [33]. Such population is at risk for hypotension due to several factors, including decreased baroreflex sensitivity, decreased ventricular diastolic filling, increased vascular stiffness, and malnutrition and dehydration [34]. Interestingly, an association between hypo- and hypertension was also found with a prevalence ranging from 13% to 32% [10]. The rising trend of hospital admissions due to hypotension among the elderly could possibly be explained by the fact that the incidence of hypertension grew with age [35].

Our study found that bed-days hypotension hospital admission accounted for 99.5% and 99.7% of the total number of hypotension hospital admissions in England and Wales, respectively. Additionally, over-night hypotension hospital admission accounted for 84.6% of the total number of hypotension hospital admissions in Australia. This reflects the preponderance of hospitalised hypotension patients who slept at least one day in hospitals in Australia, England, and Wales (length of hospital stay is more than one day). This is in line with a previous study reporting that hospital stays were longer for patients with hypotension, with the mean length of stay of seven days for symptomatic hypotension patients compared with six days for asymptomatic hypotension patients [36]. Another study found that triple lows (low minimum alveolar concentration of volatile anaesthesia, low Bispectral Index, and low blood pressure) are linked with increased length of hospital stay [37]. The length of hospital stay is one of the health metrics which is substantial to hospitals in monitoring the efficiency and quality of inpatient care. Prolonged hospital stay is associated with lower quality of care and increased adverse events, as it is risky for patients, specifically the elderly or the frail [38,39,40,41,42]. Staying for a longer period increases the possibility of hospital-acquired infection, sleep deprivation, falls, and occasionally physical and mental disequilibrium [42,43]. Multiple approaches have been developed and used to decrease the length of hospital stay. For example, as part of an orthostatic rehabilitation program among orthostatic hypotension patients, sidestepping deconditioning during bed rest and utilising pressers were used to lessen hospital stays [44]. However, previous systematic reviews have not confirmed the effectiveness of one intervention alone in consistently declining the hospital stay length in all high-risk and medically complex populations [45,46]. Policymakers, researchers, and health system leaders must cooperate to fulfil these necessities [46]. Public health policy should promote hypotension prevention strategies through various channels including healthcare professional education and social media campaigns. This should include promoting the adaptation of healthy lifestyles that involve drinking more water and less alcohol, eating small and low-carb meals, exercising regularly, avoiding strenuous activities, and sleeping with the head slightly elevated [47].

The findings of our study identified that hospital admissions for hypotension due to drugs accounted for 13.5%, 7.6%, and 10.0% of the total number of hypotension admissions in Australia, England, and Wales. This is consistent with previous studies where they found a considerable number of hospital admissions to be caused by adverse drug reactions [48,49]. For elderly patients with multiple comorbidities and who are using multiple medications, medication review for doses and potential interactions should be considered [50,51,52]. For instance, when a macrolide antibiotic is required for patients receiving calcium channel blockers, preferential use of azithromycin should be regarded as a previous study showed that the use of clarithromycin or erythromycin in older patients receiving calcium channel blockers was associated with a raised risk of shock or hypotension requiring hospital admission [53]. Furthermore, it may be sensible, for some patients, to switch from antihypertensive medications to lifestyle changes or reduce or refrain from titrating the dose upward if drug-associated hypotension occurs [1].

Our study found that hospital admission rates for drugs and other hypotension in Australia and Wales decreased among patients aged between 60–74 and 75 years and above in 2020, which could be due to several factors, including changes in prescribing practices during the COVID-19 pandemic [54,55] and changes in healthcare utilization patterns during the COVID-19 pandemic [56]. Additionally, it is important to note that the age group over 65 years is a medically heterogeneous population, and a standardized approach to medical care may not be suitable for all patients [57].

Our study found that between 1999 and 2020, there were 212,890 hypotension hospital admissions in Australia (50.8% males), 554,793 in England (50.3% males), and 29,458 in Wales (50.0% males). Still, differences in hospital admissions for each type of hypotension among males and females were remarkably few throughout our study. That is consistent with Méndez et al.’s (2018) study which found that the prevalence of orthostatic hypotension is similar among younger males and females in general, but among those aged over 75 years, the prevalence of orthostatic hypotension is decreased in females and increased in males [58].

To the best of our knowledge, this is the first study to provide the hospitalisation profile related to all types of hypotension in Australia, England, and Wales. We demonstrated the hospitalisation rates stratified by age, gender, and type with no restrictions on specific population groups. At the same time, this study has limitations. The used age categories were broad, which is not the ideal method to present them; however, this was a result of the reporting style of the public data in the medical databases used in this study. The aggregated nature of the available data in the used medical databases prevented our ability to adjust for important confounding factors that might have overestimated or underestimated our presented hospitalisation rate. Therefore, our findings should be interpreted carefully.

## 5. Conclusions

Orthostatic hypotension is the predominant type of hypotension-related hospital admissions in Australia, England, and Wales in the past 20 years and is being most pronounced in the high age groups. Future studies should aim to identify avoidable risk factors for hypotension and design effective interventions to decrease the burden of orthostatic hypotension. A public policy aimed at preventing hypotension should be promoted to reduce its associated complications.

## Figures and Tables

**Figure 1 healthcare-11-01210-f001:**
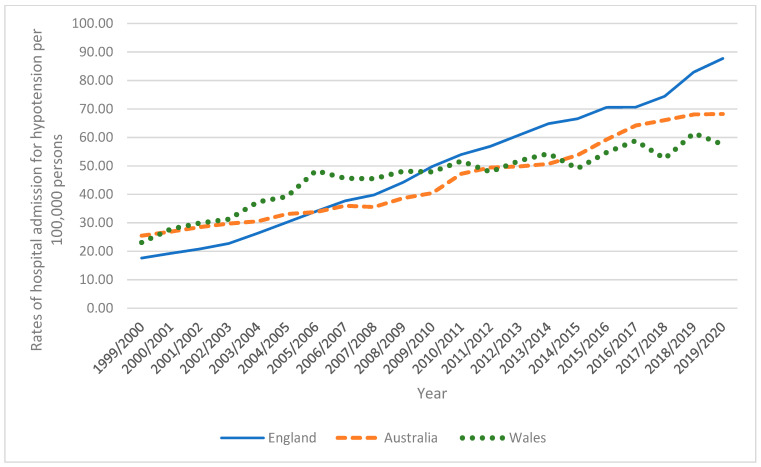
Rates of hospital admission for hypotension in England, Australia, and Wales.

**Figure 2 healthcare-11-01210-f002:**
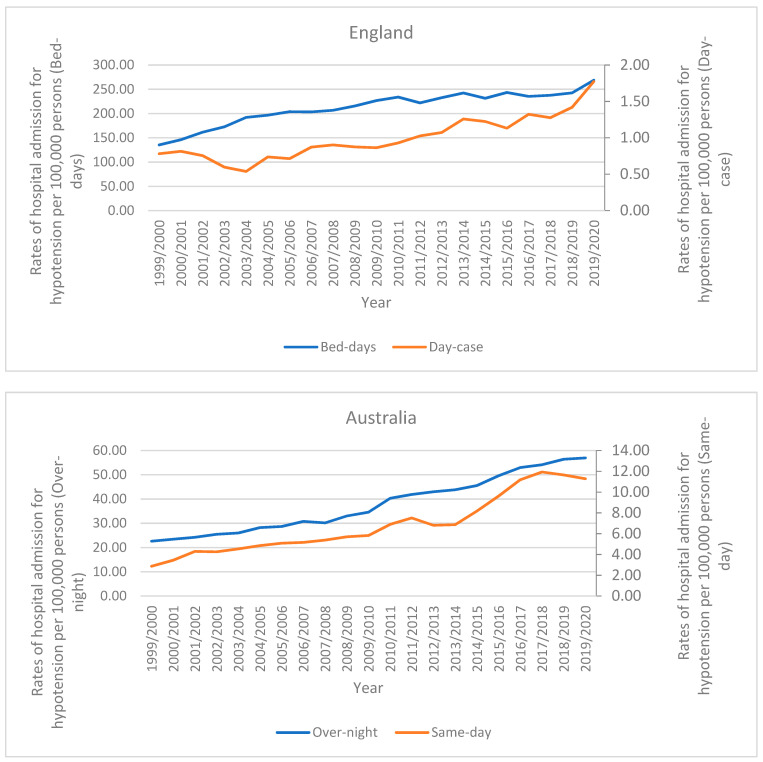
Rates of hospital admission for hypotension in England, Australia, and Wales stratified by hospital stay.

**Figure 3 healthcare-11-01210-f003:**
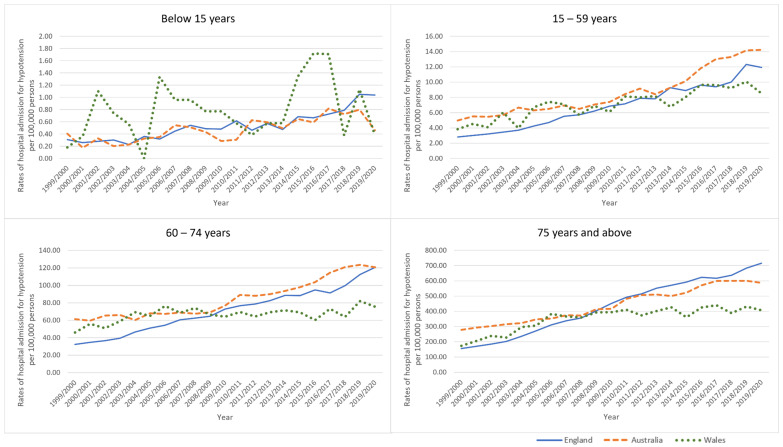
Rates of hospital admission for hypotension stratified by age group.

**Figure 4 healthcare-11-01210-f004:**
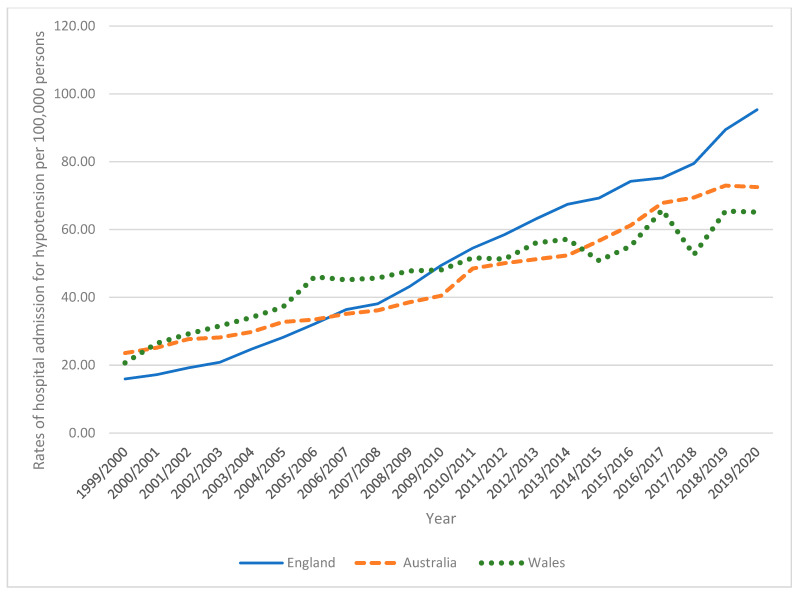
Rates of hospital admission for hypotension among males.

**Figure 5 healthcare-11-01210-f005:**
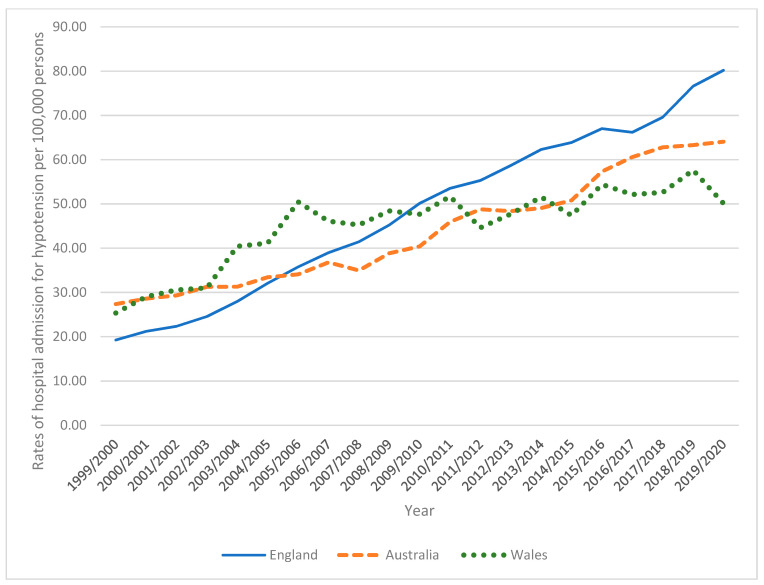
Rates of hospital admission for hypotension among females.

**Figure 6 healthcare-11-01210-f006:**
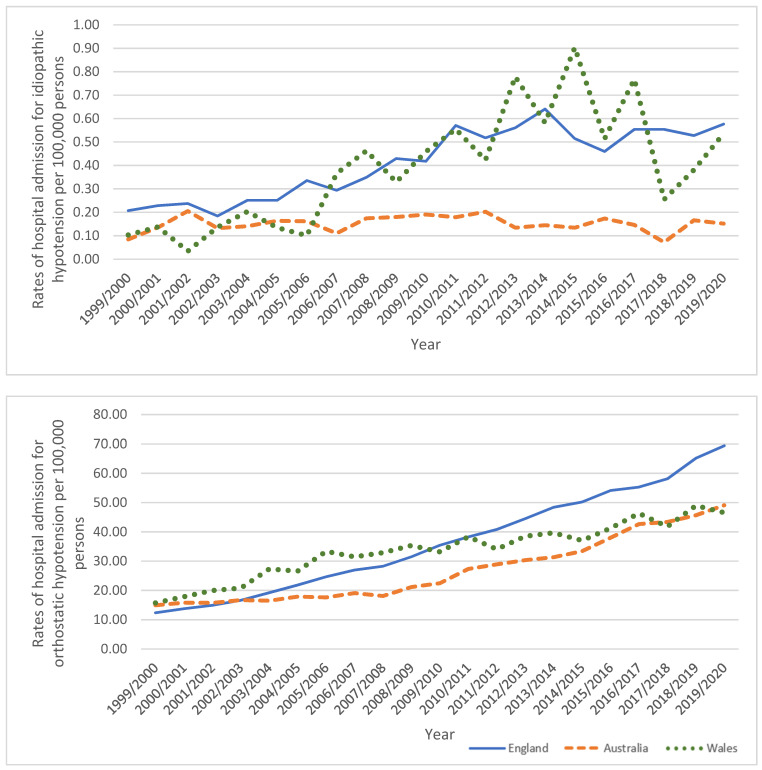
Rates of hospital admission for hypotension stratified by type.

**Figure 7 healthcare-11-01210-f007:**
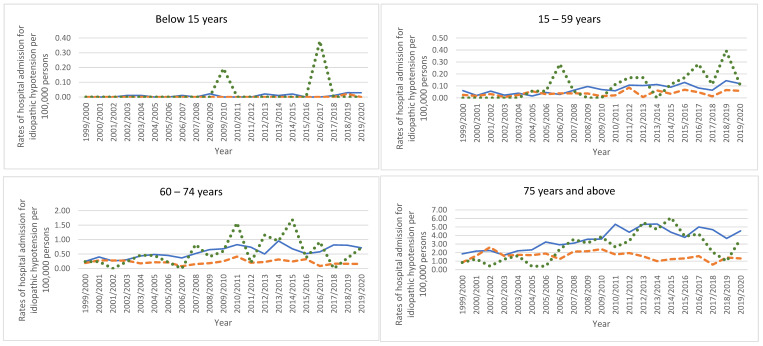
Rates of hospital admission for hypotension types stratified by age group.

**Figure 8 healthcare-11-01210-f008:**
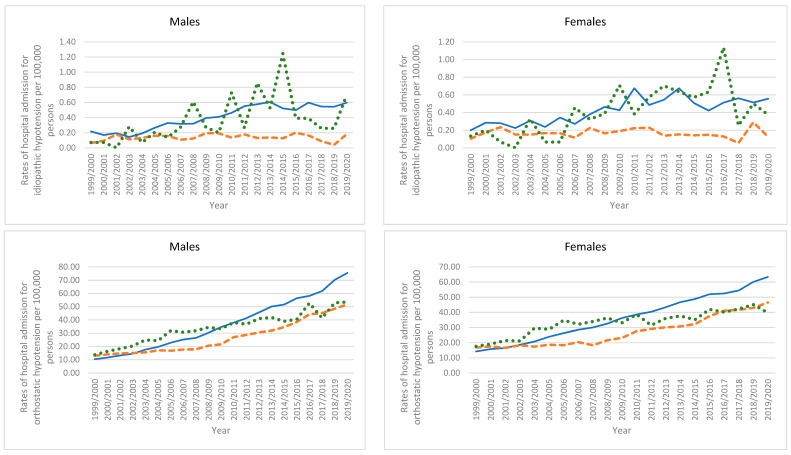
Rates of hospital admission for hypotension types stratified by gender.

**Table 1 healthcare-11-01210-t001:** Percentage of hypotension hospital admission from total number of admissions.

ICD Code	Description	Percentage from Total Number of Admissions		
		England	Wales	Australia
I95.0	“Idiopathic hypotension”	0.8%	0.9%	0.3%
I95.1	“Orthostatic hypotension”	74.7%	73.5%	61.0%
I95.2	“Hypotension due to drugs”	7.6%	10.0%	13.5%
I95.8	“Other hypotension (Postprocedural hypotension, iatrogenic, and chronic)”	0.7%	0.7%	1.4%
I95.9	“Hypotension, unspecified”	16.2%	15.0%	23.8%

**Table 2 healthcare-11-01210-t002:** Percentage of hypotension hospital admission for each age group from the total number of admissions.

Age Group	Percentage from Total Number of Admissions		
	England	Wales	Australia
Below 15 years	0.2%	0.3%	0.2%
15–59 years	8.3%	8.9%	11.9%
60–74 years	21.1%	23.4%	24.7%
75 years and above	70.4%	67.4%	63.2%

**Table 3 healthcare-11-01210-t003:** Percentage change in the hospital admission rates for hypotension from 1999–2020 stratified by age group.

	Rate of Hypotension in 1999 per 100,000 Persons (95% CI)	Rate of Hypotension in 2020 per 100,000 Persons (95% CI)	Percentage Change
Below 15 years			
England	0.31 (0.20–0.42)	1.04 (0.84–1.24)	234.8%
Wales	0.18 (−0.17–0.54)	0.38 (−0.15–0.90)	109.3%
Australia	0.41 (0.21–0.60)	0.46 (0.27–0.65)	13.5%
15–59 years			
England	2.82 (2.63–3.01)	11.93 (11.56–12.31)	323.2%
Wales	3.83 (2.90–4.76)	8.50 (7.15–9.86)	122.2%
Australia	4.98 (4.58–5.38)	14.22 (13.62–14.82)	185.5%
60–74 years			
England	32.28 (30.90–33.66)	120.66 (118.36–122.95)	273.8%
Wales	45.94 (39.45–52.44)	75.64 (68.43–82.84)	64.6%
Australia	61.35 (58.00–64.70)	120.76 (117.27–124.25)	96.8%
75 years and above			
England	155.20 (151.17–159.22)	715.37 (707.88–722.86)	360.9%
Wales	173.41 (156.70–190.12)	406.52 (384.00–429.04)	134.4%
Australia	277.99 (267.98–288.01)	586.70 (575.64–597.76)	111.0%

**Table 4 healthcare-11-01210-t004:** Percentage change in the hospital admission rates for hypotension from 1999–2020 stratified by gender.

	Rate of Hypotension in 1999 per 100,000 Persons (95% CI)	Rate of Hypotension in 2020 per 100,000 Persons (95% CI)	Percentage Change
Males			
England	15.93 (15.42–16.43)	95.33 (94.18–96.47)	498.6%
Wales	20.67 (18.29–23.04)	65.11 (61.11–69.11)	215.1%
Australia	23.55 (22.57–24.53)	72.48 (71.00–73.96)	207.8%
Females			
England	19.24 (18.70–19.78)	80.20 (79.16–81.23)	316.7%
Wales	25.35 (22.81–27.90)	50.12 (46.66–53.58)	97.7%
Australia	27.38 (26.33–28.42)	64.06 (62.69–65.44)	134.0%

**Table 5 healthcare-11-01210-t005:** Percentage change in the hospital admission rates for hypotension from 1999–2020 stratified by type.

	Rate of Hypotension in 1999 per 100,000 Persons (95% CI)	Rate of Hypotension in 2020 per 100,000 Persons (95% CI)	Percentage Change from 1999–2020
Idiopathic hypotension			
England	0.21 (0.17–0.25)	0.58 (0.51–0.64)	178.3%
Wales	0.10 (−0.01–0.22)	0.54 (0.28–0.79)	419.7%
Australia	0.08 (0.04–0.13)	0.15 (0.10–0.20)	80.5%
Orthostatic hypotension			
England	12.38 (12.07–12.69)	69.43 (68.74–70.12)	460.6%
Wales	15.76 (14.31–17.20)	46.57 (44.19–48.94)	195.6%
Australia	14.95 (14.40–15.50)	49.11 (48.25–49.96)	228.6%
Hypotension due to drugs			
England	1.54 (1.43–1.65)	4.80 (4.62–4.98)	212.7%
Wales	2.99 (2.36–3.62)	2.46 (1.91–3.01)	−17.8%
Australia	2.99 (2.74–3.24)	3.85 (3.61–4.09)	28.6%
Other hypotension			
England	0.12 (0.09–0.15)	0.67 (0.60–0.74)	470.4%
Wales	0.24 (0.06–0.42)	0.54 (0.28–0.79)	122.7%
Australia	0.85 (0.72–0.98)	0.70 (0.60–0.81)	−16.7%
Hypotension, unspecified			
England	3.38 (3.22–3.54)	12.25 (11.96–12.53)	262.3%
Wales	3.99 (3.26–4.72)	7.41 (6.47–8.36)	85.8%
Australia	6.61 (6.25–6.98)	14.43 (13.97–14.90)	118.3%

## Data Availability

Publicly available datasets were analysed in this study. This data can be found here: https://meteor.aihw.gov.au/content/394352, https://digital.nhs.uk/data-and-information/publications/statistical/hospital-admitted-patient-care-activity, and https://www.nhs.wales/ accessed on 14 January 2023.

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
