# Peer review of "Hospital Admission Due to Hypotension in Australia and in England and Wales"

_healthcare, 2023, doi:10.3390/healthcare11091210_

Round 1
Reviewer 1 Report
Let me thank you for the possibility to review this interesting research.
The authors aimed to examine the frequency of hospital admissions due to hypotension in Australia, England and Wales over past two decades as well as the main risk factors for hypotension.
Below, please, find some of my suggestions:
- line 12 - lack of the dot after "death".
- lines 45-51 - I would suggest supplementing this paragraph with a brief information about the possible primary causes of the shock.
- lines 54-56 - All antihypertensive drugs can cause hypotension, so listing all groups seems risky to me. I would limit this sentence to statement that antihypertensive drugs, taken in excess or off-label, can cause a significant reduction in blood pressure.
- line 56 - the sentence "Hypotension is also associated 56 with certain diseases such as Parkinson’s disease, type-1 diabetes, and carotid artery stenosis or 57 may result from iatrogenic complications while the patient admitted to the hospital or 58 maybe the reason for the hospital admission" seems very unclear to me, please change.
- line 67 - shoud be "death" instead of die.
- line 71 - "provide evidence to health care providers" - please change.
- line 106 - is instead of are.
- lines 272 - 277 - In my opinion, this paragraph should be moved to "Introduction". The order of the causes of hypotension should be changed, listing emotional factors at the end of the list rather than at the beginning.
Best wishes.
Author Response
Reviewer 1:
Let me thank you for the possibility to review this interesting research.
The authors aimed to examine the frequency of hospital admissions due to hypotension in Australia, England and Wales over past two decades as well as the main risk factors for hypotension.
Below, please, find some of my suggestions:
- line 12 - lack of the dot after "death".
- Thank you for this comment, we have now addressed this comment.
- lines 45-51 - I would suggest supplementing this paragraph with a brief information about the possible primary causes of the shock.
- Thank you for this comment, we have now addressed this comment, see page 3, lines 65-66.
- lines 54-56 - All antihypertensive drugs can cause hypotension, so listing all groups seems risky to me. I would limit this sentence to statement that antihypertensive drugs, taken in excess or off-label, can cause a significant reduction in blood pressure.
- Thank you for this comment, we have now addressed this comment, see pages 3 and 4, lines 72-74.
- line 56 - the sentence "Hypotension is also associated 56 with certain diseases such as Parkinson’s disease, type-1 diabetes, and carotid artery stenosis or 57 may result from iatrogenic complications while the patient admitted to the hospital or 58 maybe the reason for the hospital admission" seems very unclear to me, please change.
- Thank you for this comment, we have now addressed this comment and rephrased the sentence in page 4, lines 74-78.
- line 67 - shoud be "death" instead of die.
- Thank you for this comment, we have now addressed this comment, see page 4, line 86.
- line 71 - "provide evidence to health care providers" - please change.
- Thank you for this comment, we have now addressed this comment and removed this sentence to avoid confusion, see page 4, lines 89-90.
- line 106 - is instead of are.
- Thank you for this comment, we have now addressed this comment, see page 5, line 126.
- lines 272 - 277 - In my opinion, this paragraph should be moved to "Introduction". The order of the causes of hypotension should be changed, listing emotional factors at the end of the list rather than at the beginning.
- Thank you for this comment, we have now addressed this comment and re-allocated the order of the causes, see page 28 and 29. However, we prefer to keep it in the discussion as a possible interpretation for the causes of increased admission rate for the reader.
Best wishes.
Reviewer 2 Report
Thank you for allowing me to review this manuscript on a very interesting and frequent topic. My recommendation would be that in the discussion and conclusions the strategy or strategies that the author would propose for this entity to be prevented and diminished from the public policy point of view be written down.
Author Response
Reviewer 2:
Thank you for allowing me to review this manuscript on a very interesting and frequent topic. My recommendation would be that in the discussion and conclusions the strategy or strategies that the author would propose for this entity to be prevented and diminished from the public policy point of view be written down.
- Thank you for this comment, we have now addressed this comment in the discussion and in the conclusion section, lines 411-415.
Reviewer 3 Report
This is a solid paper on a topic that is clinically important and underrecognized. The main problem is the data calculation and presentation.
The authors repeatedly use the term "-fold" but it was not computed correctly. The -fold term refers to a number calculated by dividing the two numbers: new/original. These authors did not do that. When using the percentage (%) change, the math is performed as follows: (New-original)/original X 100.
All the -fold terms have to be recomputed.
Example "2.62-fold from 4,848 in 1999 to 17,533 in 2020" This is incorrect. This change is 3.6-fold!
In addition, certain paragraphs use -fold and percentage change computations. See lines 154-157, for example. Also lines 178-180 as well. Both computations appear in one sentence! This is much too confusing to the reader. Consider presenting all numbers as percentage increase or decrease, especially since several of the Tables present data this way; e.g., Tables 3 and 4.
The age group “15-59” is used throughout. This is not appropriate medically. The difference between a teenager and a 59 year old cannot be ignored. Need finer division, perhaps: 15-25, 25-40, 40-59.
Author Response
Reviewer 3:
This is a solid paper on a topic that is clinically important and underrecognized. The main problem is the data calculation and presentation.
The authors repeatedly use the term "-fold" but it was not computed correctly. The -fold term refers to a number calculated by dividing the two numbers: new/original. These authors did not do that. When using the percentage (%) change, the math is performed as follows: (New-original)/original X 100.
All the -fold terms have to be recomputed.
- Thank you for this comment, we have now addressed this comment and removed the term folds and presented the percentage change instead throughout the manuscript.
Example "2.62-fold from 4,848 in 1999 to 17,533 in 2020" This is incorrect. This change is 3.6-fold!
In addition, certain paragraphs use -fold and percentage change computations. See lines 154-157, for example. Also lines 178-180 as well. Both computations appear in one sentence! This is much too confusing to the reader. Consider presenting all numbers as percentage increase or decrease, especially since several of the Tables present data this way; e.g., Tables 3 and 4.
The age group “15-59” is used throughout. This is not appropriate medically. The difference between a teenager and a 59 year old cannot be ignored. Need finer division, perhaps: 15-25, 25-40, 40-59.
- Thank you for this comment. Unfortunately, the data available in the medical databases are reported using four age group categories (below 15 years, 15-59 years, 60-74 years, and 75 years and older). However, based on the reviewer comment, we have now mentioned this point in the study limitations section, see page 32, line 446-448.
Reviewer 4 Report
The manuscript is well-written, covering all the aspects of a nice scientific research paper. Although a few changes would help.
Sice the data is representing different places, a table showing exact details would be easier for the reader to understand.
Apart from this, it has a lot of things in excess and repetitions. So would suggest to be more direct regarding the topic.
Overall, it is a nice read.
Author Response
Reviewer 4:
The manuscript is well-written, covering all the aspects of a nice scientific research paper. Although a few changes would help.
Sice the data is representing different places, a table showing exact details would be easier for the reader to understand.
Apart from this, it has a lot of things in excess and repetitions. So would suggest to be more direct regarding the topic.
Overall, it is a nice read.
- Thank you for this comment. As we were presenting the data across the three countries we were forced to repeat some sentences upon stating the comparison across different study findings. However, as the reviewer suggested Tables 1, 2,3,4, and 5 presents the exact details of our findings.
Reviewer 5 Report
The study by Sara Ibrahim Hemmo et al. analyzed hospital admission due to hypotension in Australia, England, and Wales. The authors concluded that in the past two decades, orthostatic hypotension was the most prevalent type of hypotension that required hospitalization in Australia, England, and Wales. Age was identified as the primary risk factor for hypotension across all causes. Overall speaking, the analysis is comprehensive, and the research is interesting. I have some concerns:
1. The authors should explain the definition of the specific words in the manuscript, such as "bed-days hypotension" and "over-night hypotension".
2. What is the novel finding of this study compared to previous studies?
3. Based on this study, what is the potential new indication for preventing or treating hypotension patients in the clinic?
Author Response
Reviewer 5:
The study by Sara Ibrahim Hemmo et al. analyzed hospital admission due to hypotension in Australia, England, and Wales. The authors concluded that in the past two decades, orthostatic hypotension was the most prevalent type of hypotension that required hospitalization in Australia, England, and Wales. Age was identified as the primary risk factor for hypotension across all causes. Overall speaking, the analysis is comprehensive, and the research is interesting. I have some concerns:
- The authors should explain the definition of the specific words in the manuscript, such as "bed-days hypotension" and "over-night hypotension".
- Thank you for this comment, we have now added the explanation in the method section, see pages 6 and 7, lines 140-143.
- What is the novel finding of this study compared to previous studies?
- Thank you for this comment. To the best of our knowledge, this is the first study to examine trends of admissions in Australia, England and Wales during the past 20 years without restricting the study population to specific age group or gender or type of hypotension. This is highlighted in the last paragraph in the discussion section, see page 32, lines 443-446.
- Based on this study, what is the potential new indication for preventing or treating hypotension patients in the clinic?
- Thank you for this comment. We have now added recommendation to health policy in the discussion section as the following “Public health policy should promote hypotension prevention strategies through various channels including healthcare professional education and social media campaigns. This should include promoting the adaptation of healthy lifestyle that involve drinking more water and less alcohol, eating small and low-carb meals, exercise regularly, avoiding strenuous activities, and sleeping with bed head slightly elevated”, see lines 411-415.
Round 2
Reviewer 3 Report
Much improved.